# [Re]: Value Alignment Verification

## Reproducibility Summary

**Scope of Reproducibility**

The main goal of the paper "Value Alignment Verification" [7] is to test the alignment of a robot's behavior efficiently with human expectations by constructing a minimal set of questions. To accomplish this, the authors propose algorithms and heuristics to create the above questionnaire. They choose a wide range of gridworld environments and a continuous autonomous driving domain to validate their put forth claims. We explore value alignment verification for gridworlds incorporating a non-linear feature reward mapping as well as an extended action space.

**Methodology**

We re-implemented the pipeline with Python using mathematical libraries such as Numpy and Scipy. We spent approximately two months reproducing the targeted claims in the paper with the first month aimed at reproducing the results for algorithms and heuristics for exact value alignment verification. The second month focused on extending the action space, additional experiments, and refining the structure of our code. Since our experiments were not computationally expensive, we carried out the experiments on CPU. The code is available at https://anonymous.4open.science/r/vavrc21/.

**Results**

The techniques proposed by authors in [7] can successfully address the value alignment verification problem in different settings. We empirically demonstrate the effectiveness of their proposals by performing exhaustive experiments with several variations to their original claims. We show high accuracy and low false positive and false negative rates in the value alignment verification task with a minimum number of questions for different algorithms and heuristics.

**What was easy**

The problem statement, as well as the implementation of algorithms and heuristics, were straightforward. We also took aid from the original repository published with the paper. However, we implemented the entire pipeline from scratch and incorporated several variations to our code to perform additional designed experiments.

**What was difficult**

Comprehending different algorithms and heuristics proposed in prior works along with their mathematical formulation and reasoning for their success in the given task was considerably difficult. Additionally, the original code base had several redundant files, which created initial confusion. We iterated and discussed the arguments in the paper and prior work several times to thoroughly understand the pipeline. Nevertheless, once the basics were clear, the implementation was comparatively simple.

**Communication with original authors**

We reached out to the authors numerous times via email to seek clarifications and additional implementation details. The authors were incredibly receptive to our inquiries, and we appreciate their thorough and prompt responses.

Submitted to ML Reproducibility Challenge 2021. Do not distribute.

## 1  Introduction

Autonomous agents are used for complex, challenging, riskier, and dangerous tasks which brings up the need of *verifying* whether the agents act in a way that is both optimal and safe *w.r.t* another agent that has already been performing the said task (example, a human agent). This problem of verifying the *alignment* of one agent's behavior *w.r.t* another agent is known as *Value Alignment Verification*. The original paper [7] proposes a framework for efficient value alignment verification. They discuss three different settings of increasing difficulty in terms of verification:

1. *explicit human, explicit robot*: where both the agents are completely aware of their reward functions.
2. *explicit human, implicit robot*: where the human agent is aware of its reward function but the robot agent can only be queried about its action preferences on different states.
3. *implicit human, implicit robot*: where the only basis of value alignment is through preferences over trajectories.

Depending on the setting, value alignment can be either exact or approximate. We try to reproduce and validate the results for the proposed framework on the first and second setting, i.e., (*explicit human, explicit robot*) and (*explicit human, implicit robot*). The experiments involve gridworld environments with a deterministic action space. The aim of value alignment verification is to create a *questionnaire* using the human agent's knowledge (reward function or trajectory preferences) that can be given to any agent in order to verify alignment. Efficient verification aims to minimize the number of queries in the *questionnaire*. While few works on value alignment discuss qualitative evaluation of trust [10] or asymptotic alignment of an agent's performance via interactions and active learning [9] [8] [13], [7] solely focuses on verifying value alignment for two or more agents with a learned policy. The objective is to efficiently test compatibility of different robots with human agents. In the following sections, we reiterate the formal definition of value alignment as stated by the original authors (Value Alignment Verification in Section 3 and Exact Value Alignment Verification in Section 4), followed by our experiment settings in Section 7 and subsequent observations in Section 8.

## 2  Notation

We use the notation proposed in [2], where a Markov Decision Process (MDP) $M$ is defined by an environment $E$ and a reward function $R$. An environment $E = (S, A, P, S_0, \gamma)$ where $S$ is a set of states, $A$ is a set of actions, $P$ is a transition function, $P : S \times A \times S \to [0, 1]$, $\gamma \in [0, 1)$ is discount factor and a distribution over initial states $S_0$. The reward function $R : S \to \mathbb{R}$. A policy $\pi : S \times A \to [0, 1]$ from states to a distribution over actions. The state and state-action values of a policy $\pi$ are $V_R^\pi(s) = \mathbb{E}_\pi[\sum_{t=0}^\infty \gamma^t R(s_t)|s_0 = s]$ and $Q_R^\pi(s, a) = \mathbb{E}_\pi[\sum_{t=0}^\infty \gamma^t R(s_t)|s_0 = s, a_0 = a]$ for $s \in S$ and $a \in A$. The optimal value functions are, $V_R^*(s) = \max_\pi V_R^\pi(s)$ and $Q_R^*(s, a) = \max_\pi Q_R^\pi(s, a)$. Let $A_R(s) = argmax_{a' \in A} Q_R^*(s, a')$ denote the set of optimal actions at a state $s$ under the reward function $R$. Then $A_R(s) = \{a \in A | \pi_R^*(a|s) > 0\}$. It is assumed that reward function is linear under state features ([14], [3], [4]) $\phi : S \to \mathbb{R}^k$, such that $R(s) = \mathbf{w}^T \phi(s)$, where $\mathbf{w} \in \mathbb{R}^k$. Note that there is no restriction on the features $\phi$, therefore these features could be complex non-linear functions of the state as well. The state-action value function can be written in terms of features ([1]) as $Q_R^\pi(s, a) = \mathbf{w}^T \Phi_\pi^{(s,a)}$ where $\Phi_\pi^{(s,a)} = \mathbb{E}_\pi[\sum_{t=0}^\infty \gamma^t \phi(s_t)|s_0 = s, a_0 = a]$.

## 3  Value Alignment Verification

Consider two agents (for instance, a human and a robot) where the first agent's (human) reward function provides the ground truth for the value alignment verification of the second agent (robot). The definition is as follows:

**Definition 1** *Given reward function R, a policy $\pi'$ is $\epsilon$-value aligned in environment E if and only if*

$$V_R^*(s) - V_R^{\pi'}(s) \le \epsilon, \forall s \in S \tag{1}$$

The aim of the study [7] is **efficient value alignment verification** which, *formally*, is a solution for the following:

$$\min_{T \subseteq \mathcal{T}} |T|, \text{ s.t. } \forall \pi' \in \Pi, \forall s \in S$$

$$V_R^*(s) - V_R^{\pi'}(s) > \epsilon \implies Pr[\pi' \text{ passes test } T] \le \delta_{\text{fpr}} \tag{2}$$

$$V_R^*(s) - V_R^{\pi'}(s) \le \epsilon \implies Pr[\pi' \text{ fails test } T] \le \delta_{\text{fnr}}$$

where $\mathcal{T}$ is the set of all possible queries, $\Pi$ is set of all policies for which the test is designed, $\delta_{fnr}, \delta_{fpr} \in [0, 1]$ are the false negative and false positive rates, and $|T|$ is the size of test $T$. When $\epsilon = \delta_{fpr} = 0$, the authors call this setting *exact value alignment verification*.

## 4   Exact Value Alignment Verification

Exact value alignment verification is not possible, even for finite MDPs, when we can only query the robot agent for its action preferences. Therefore, it is possible only in the most idealized setting, i.e., *explicit human, explicit robot*.

**Definition 2** *Define an agent $\pi'$ to be rational ([12]) if:*

$$\forall a \in A, \pi'(a|s) > 0 \implies a \in argmax_a Q^*_{R'}(s, a) \tag{3}$$

*where $argmax_a Q^*_{R'}(s, a)$ is the optimal state-action value function for the reward function $R'$.*

As there exist infinitely many reward functions which can return the same optimal policy ([11]), determining that $\exists s \in S, R(s) \neq R'(s)$ does not necessarily imply that agents with the reward functions $R, R'$ are not aligned. We provide an example of this in Figure 1, where the optimal policy for human and robot is the same; thus, they are aligned. However, the rewards are different, as mentioned in Table 1.

Figure 1: Counterexample with same optimal policy for human and robot

Table 1: Human and robot rewards for gridworld (Figure 1)

| State Color | Terminal State | Human Reward | Robot Reward |
|---|---|---|---|
| Blue | No | - 0.6157 | - 0.5316 |
| White | No | - 0.3107 | - 0.0694 |
| Green | Yes | + 0.7242 | + 0.8441 |

**Definition 3** *Define the set of all the optimal policies under the reward function R as OPT(R).*

$$OPT(R) = \{\pi | \pi(a|s) > 0 \implies a \in argmax_a Q^*_R(s, a)\}$$

Looking at Definition 1 and Equation 3 simultaneously makes it evident that for a rational robot, if all of its optimal policies are also optimal under ground truth reward function $R$; the robot is exactly aligned with the human.

**Corollary 1** *We have **exact value alignment** in environment E between a rational robot with reward function $R'$ and a human with reward function R if $OPT(R') \subseteq OPT(R)$.*

Revisiting the inspiration ([11]) of the original author's proposed approach for efficient exact value alignment -

**Definition 4** *Given an environment E, the **consistent reward set** (CRS) of a policy $\pi$ in environment E is defined as the set of reward functions under which $\pi$ is optimal*

$$CRS(\pi) = \{R | \pi \in OPT(R)\} \tag{4}$$

When $R(s) = \mathbf{w}^T \phi$, the *CRS* is of the form ([11], [6]):

**Corollary 2** *Given an environment E, the $CRS(\pi)$ is given by the following intersection of half-spaces:*

$$\{\mathbf{w} \in \mathbb{R}^k | \mathbf{w}^T (\Phi_\pi^{(s,a)} - \Phi_\pi^{(s,b)}) \geq 0, \forall a \in argmax_{a' \in A} Q^\pi_R(s, a'), b \in A, s \in S\}$$

Since the boundaries of the *CRS* polytope is consistent with a policy that may not be aligned with optimal policy (e.g. zero reward), we remove all such boundary cases to obtain a modified set called *aligned reward polytope* (*ARP*).

## 5 Reproducing Exact Value Alignment

In this section, we explain the procedure in order to verify the claims made in the paper regarding sufficient conditions for provable verification of exact value alignment (explained in Section 4). We verify exact value alignment in disparate settings proposed by the authors for *explicit human - explicit robot* setting. If we have access to the value or reward function of a human, we term it as *explicit human*. A similar notion is applicable for the robot as well.

**Theorem 1** *Under the assumption of a rational robot (defined in Section 4) that shares linear reward features with the human, efficient exact value alignment verification is possible in the following query settings: (1) Query access to reward function weights $w^{'}$, (2) Query access to samples of the reward function $R^{'}(s)$, (3) Query access to $V_{R'}^{*}(s)$ and $Q_{R'}^{*}(s, a)$, and (4) Query access to preferences over trajectories.*

**Case 1** *Reward Weight Queries*

A brute-force paradigm can be implemented to evaluate an explicit robot optimal policy under the human reward function. However, there exists another succinct verification test. We need to query the weight vector $w^{'}$ of the robot (here, $R^{'}(s) = (w^{'})^T \phi(s)$, $\phi(s)$ is the feature vector of state $s$). The paper asserts that it is possible to form a test (defined later as $\Delta$) that uses the obtained $w^{'}$ to verify alignment. Additionally, this query to the weight vector $w^{'}$ is done in constant time, and the test is linear in the number of questions.

**Definition 5** *Given an MDP M composed of environment E and reward function R, the aligned reward set (ARS) is defined as the following set of reward functions:*

$$ARS(R) = \{R^{'} | OPT(R^{'}) \subseteq OPT(R)\}$$

We state the lemma which proves the sufficient condition for exact value alignment and direct the interested readers for the proof of the lemma to refer the paper.

**Lemma 1** *Given an MDP M = (E, R), the human's and robot's reward function R and $R^{'}$ respectively can be represented as linear combinations of features $\phi(s) \in R^k$, i.e., $R(s) = w^T \phi(s)$, $R^{'}(s) = w^{'T}\phi(s)$, and given an optimal policy $\pi_R^*$ under R, we have*

$$w^{'} \in \cap_{(s,a,b) \in \mathcal{O}} \mathcal{H}_{s,a,b}^R \implies R^{'} \in ARS(R)$$

*where*

$$\mathcal{H}_{s,a,b}^R = w | w^T(\Phi_\pi^{(s,a)}) - \Phi_\pi^{(s,b)}) > 0 \text{ and } \mathcal{O} = \{(s,a,b) | s \in S, a \in \mathcal{A}_R(s), b \neq \mathcal{A}_R(s)\}$$

**Definition 6** *The intersection of half-spaces $\left(\cap_{(s,a,b) \in \mathcal{O}} \mathcal{H}_{s,a,b}^R\right)$ is defined as the Aligned Reward Polytope (ARP). The design of ARP in the form of $\Delta$ matrix is defined as follows:*

$$\Delta = \begin{bmatrix} \Phi_\pi^{(s,a)}) - \Phi_\pi^{(s,b)} \\ \vdots \end{bmatrix}$$

In the above equation, $a$ is an optimal action at state $s$, and $b$ is a non-optimal action. The actions in the trajectory following $a$ and $b$ are optimal. Each row of $\Delta$ represents the normal vector for a strict half-space constraint based on feature count differences between an optimal and sub-optimal action. Therefore, for a robot weight vector $w^{'}$, if $\Delta w^{'} > 0$, the robot is aligned. We follow the steps mentioned in the original paper to include only non-redundant half-space normal vectors in $\Delta$. We enumerate all possible half-space normal vectors corresponding to each state $s$, optimal action $a$, and non-optimal action $b$. We accumulate only non-redundant half-space normal vectors:

1. **Removal of Duplicate Vectors:** To remove *duplicate vectors*, we compute the cosine distance between the half-space normal vectors. One vector in each of the pairs of vectors with cosine distance within a small precision value (we select 0.0001) is retained in $\Delta$, others being discarded. All zero vectors are also removed.

2. **Removal of Redundant Vectors:** According to the paper, the set of *redundant vectors* can be found efficiently using the *Linear Programming* approach. To check if a constraint $a^T x \leq b$ is necessary, we first remove that constraint and solve the linear programming problem. If the optimal solution is still constrained to be less than or equal to b, that constraint can be safely discarded. After removing all such redundant vectors, we get only a set of non-redundant half-space normal vectors.

**Case 2** *Reward Queries*

In this case, the tester seeks for the rewards of the robot. Here, a tester is same as a user (human) who wishes to verify the alignment of a robot. Since it is assumed that both human and robot have access to their state feature vectors, and from the equation $R(s) = w^T \phi(s)$, we obtain the weight vector for the robot, and this case reduces to Case 1. Let $\Phi_M$ be defined as the matrix where each row corresponds to the feature vector $\phi(s)^T$ for a distinct state $s \in S$. In order to solve the system of linear equation for obtaining the weight vector, the number of queries needed is $rank(\Phi_M)$.

**Case 3** *Value Function Queries*

The tester seeks the action value function and the value function for each state in this case setting. Subsequently, the reward weights for the robot are obtained with the aid of the following equations:

$$R^{'}(s) = (w^{'})^T x \text{ and } R^{'}(s) = Q_{R'}^*(s,a) - \gamma \mathbb{E}_{s'}[V_{R'}^*(s^{'})]$$

This case also boils down to Case 1 as we obtain the weight vector for the robot. According to the paper, if we define the maximum degree of the MDP transition function as

$$d_{\max} = \max_{s \in S, a \in A} |\{s^{'} \in S | P(s,a,s^{'}) > 0\}|,$$

then at most $d_{max}$ possible next state queries are needed to evaluate the expectation. Therefore, at most $rank(\Phi_M)(d_{max}+1)$ queries are required to recover the robot's weight vector.

**Case 4** *Preference Queries*

We obtain preference over trajectories $\xi$ as judged by the human. Each preference $\xi_A > \xi_B$, induces a constraint $(w^{'})^T(\Phi(\xi_A) - \Phi(\xi_B)) > 0$, where $\Phi(\xi) = \sum_{i=1}^n \gamma^i \phi(s_i)$ is the cumulative discounted reward features (linear combination of state features) along a trajectory. Therefore, we construct $\Delta$ where each row corresponds to a half-space normal resulting from preference over individual trajectories. In this case, only a logarithmic number of trajectories are needed from all possible trajectory space to obtain $\Delta$ matrix and proceed to verify alignment of robot. We obtain all valid trajectories, perform preprocessing (remove duplicate & redundant vectors), and observe that the total number of queries is bounded by logarithmic number of trajectories we started with ([5]).

# 6 Value Alignment Verification Heuristics

When the robot acts as a black box and can provide state action preferences instead of a policy, the authors propose three heuristics; *Critical States*, *Machine Teaching* and *ARP Heuristic*. Each heuristic consists of a method for selecting the states at which the robot is tested and queries for an action, subsequently checking if the action is optimal under human's reward function. It is important to note that for these heuristics, $\delta_{fpr} > 0$, as there is no guarantee for the robot to always take the same action at a given state.

1. **Critical States Heuristic:** Inspired by the notion of *critical states* (CS) [10], the heuristic test consists of states for which $Q_R^*(s, \pi_R^*(s)) - \frac{1}{|A|} \sum_{a \in A} Q_R^*(s,a) > t$, where t is a threshold value. This intuitively states the importance of a particular state and tends to make the verification efficient.

2. **Machine Teaching Heuristic:** This heuristic is based on Set Cover Optimal Teaching (SCOT) [6], which approximates the minimal set of state-action trajectories necessary to teach a specific reward function to an IRL agent. [6] show that in the intersection of half-spaces that define the CRS (Corollary 2), the learner recovers a reward function. The authors use SCOT to create informative trajectories and create alignment tests by seeking a robot action at each state along the trajectory. Producing a test with SCOT takes longer than CS heuristic, but unlike CS, SCOT prevents repetitive inquiries by reasoning about reward features over a set of trajectories.

3. **ARP Heuristic:** This heuristic is a black-box alignment heuristic (ARP-bb) based on the ARP definition. ARP-bb first computes $\Delta$, then uses linear programming to remove duplicate half-space constraints, subsequently asks for robot actions from the states corresponding to the non-redundant constraints (rows) in $\Delta$. Intuitively, the states probed by ARP-bb are significant because different actions disclose vital information about the reward function. ARP-bb approximates testing each half-space constraint by using single-state action queries. As a result, ARP-bb trades off increased approximation error in exchange for a lower query and computational complexity.

# 7 Experiments

In this section, we describe several experiments carried out in order to investigate the following:

1. **Algorithms and Heuristics**: Comparison of different algorithms and heuristics in different gridworlds. We tabulate the performance of testers (accuracy, false positive rate, false negative rate, and the number of queries presented to the robot for verification) w.r.t different gridworld widths ranging from 4 to 8 and feature size from 3 to 8. The dimension of feature for a state is termed as number of features or feature size. Our experiments confine these state features $\phi$ to be one-hot vectors only.

2. **Diagonal Actions**: Comparison of algorithms and heuristics in gridworlds with an extended action space. We allow diagonal movement between standard movements. This increases the standard 4 actions (left, up, right, and down) to 8 actions (left-up-diagonal, up-right-diagonal, right-down-diagonal, and down-left-diagonal). Again, we tabulate the performance of testers w.r.t different gridworld widths.

3. **Non-linear reward and state-feature relationships**: Comparison of different algorithms and heuristics with non-linear (*cubic* and *exponential*) reward $R$ and state-feature $\phi(s)$ relationships. In *cubic*, we approximate the linear behavior when $w^T \phi(s) \approx 0$, else not. The exact relationship we consider is $R = x^3 + 10x$ where $x = w^T \phi(s)$. In *exponential*, we completely remove the linear relationship between $R$ and $\phi(s)$ and consider $R = e^{w^T \phi(s)}$. We tabulate the performance of testers w.r.t different gridworld widths in both cases.

4. **Critical States Tester for different thresholds**: Comparison of Critical States Tester performance with different threshold values (0.0001, 0.2 and, 0.8) for a state to be critical.

Section 8 provides the results for one algorithm (Reward Weight Tester) and one heuristic (Critical State Tester) and plots relevant to their accuracy and number of test queries. We redirect readers to Section 2 of Supplementary Material for the detailed tabulated performance of all algorithms, heuristics, and the plots related to false positive and false negative rates. Also, note that the default gridworld rows are 4, gridworld width is 8, number of actions is 4, feature size is 5, reward and state-feature relationship is linear ($R = w^T \phi(s)$), and threshold value of Critical States Tester is 0.2.

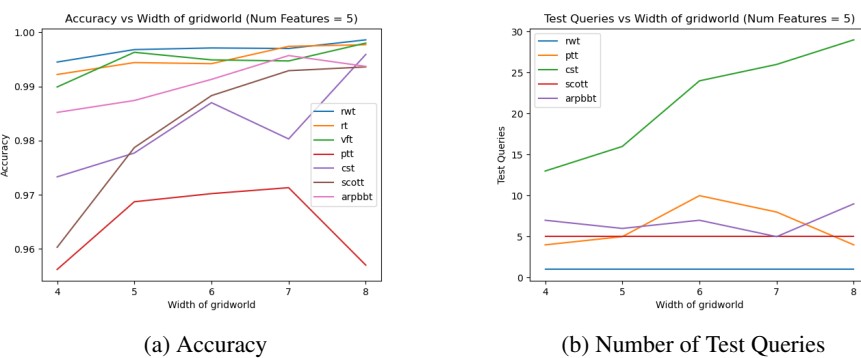

| (a) Accuracy | (b) Number of Test Queries |

Figure 2: Tester performance for different gridworld widths (num features = 5)

We created 100 different human agents for each experiment, and for each human agent, we created 100 different robots to check their alignment. Each human agent corresponds to a different human weight vector whose each element is sampled from a normal distribution with mean 0 and variance 1. Different robot agents correspond to different robot weights that are obtained by adding a random normal noise vector to the corresponding human weight vector. The elements of the noise vector are sampled from the same normal distribution. Further, we normalize the robot and human weight vector to have a unit norm. In total, we run 1.32 million experiments to address the points mentioned above.

# 8 Results

In the plots and following discussion, `rwt` indicates Reward Weight Queries Tester, `rt` indicates Reward Queries Tester, `vft` indicates Value Function Queries Tester, `ptt` indicates Preference Trajectory Queries Tester, `cst` indicates Critical States Tester, `scott` indicates SCOT Tester, and `arpbbt` indicates ARP Black Box Tester.

## 8.1 Algorithms and Heuristics

The comparison between the performance of different algorithms and heuristics is presented in Table 2, and Figure 2 (for different gridworld widths), Table 3 and Figure 3 (for different feature sizes). The plots obtained are similar to the plots presented in [7]. We averaged the accuracy over 10000 experiments (100 different human agents and 100 different robots corresponding to each human agent) and round up to 3 decimal places. We notice that `scott` takes the maximum time to verify 100 different robots whereas `rwt` takes the minimum time. The details are present in Section 2 of the Supplementary Material. We observe that, in general, the algorithms for exact value alignment verification have slightly higher accuracy. We also observe that the accuracies and number of test queries increase with increasing feature sizes. Note that, in [7], the accuracy in various plots is considered as (1 - false positive rate), while we have different plots for both. We attribute the comparatively low accuracy with `ptt` to comparatively bad trajectory queries.

Table 2: Different testers versus gridworld widths

| Tester | Width | Accuracy | False positive rate | False negative rate | Number of queries |
|---|---|---|---|---|---|
| rwt | 4 | $0.995 \pm 0.013$ | $0.005 \pm 0.011$ | $0.001 \pm 0.005$ | 1 |
| | 6 | $0.997 \pm 0.007$ | $0.002 \pm 0.005$ | $0.001 \pm 0.005$ | 1 |
| | 8 | $0.999 \pm 0.004$ | $0.001 \pm 0.004$ | $0.000 \pm 0.002$ | 1 |
| cst | 4 | $0.973 \pm 0.043$ | $0.000 \pm 0.000$ | $0.027 \pm 0.043$ | 13 |
| | 6 | $0.987 \pm 0.018$ | $0.000 \pm 0.000$ | $0.013 \pm 0.018$ | 24 |
| | 8 | $0.996 \pm 0.007$ | $0.000 \pm 0.001$ | $0.004 \pm 0.007$ | 29 |

As per Definition 1, we require $\delta_{fpr} = \epsilon = 0$ for Exact Value Alignment Verification; hence false negatives can be present in the corresponding algorithms. Further, we discussed with the authors the possibility of false positives in these algorithms, and we concluded that since we do not consider all possible trajectories in a gridworld (which is exponential in the number of actions), false positives can be present. However, we observe that both false positive and false negative rates are negligibly small. These results empirically show that indeed the proposed algorithms and heuristics successfully identify the alignment between human agents and robots.

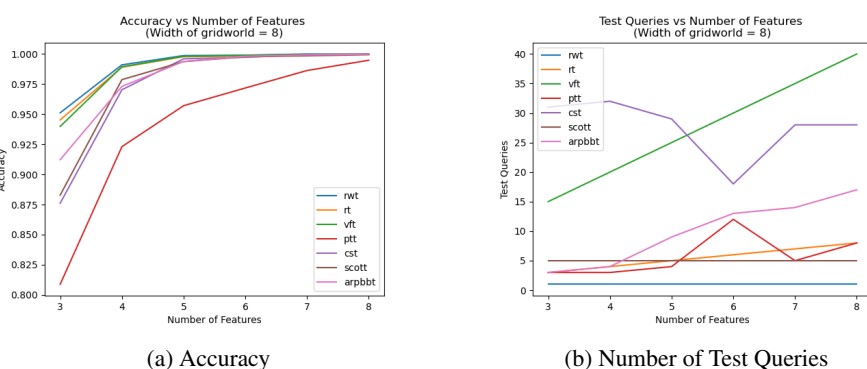

(a) Accuracy                    (b) Number of Test Queries

Figure 3: Tester performance for different number of features

In Figure 2b, the number of queries indicates the size of the questionnaire, i.e., $|T|$. The total number of queries required to verify the value alignment with `cst` is higher than other heuristics owing to its simpler mechanism for obtaining state queries. We observe that `arpbbt` is also bounded by the logarithm of the total number of queries, i.e., trajectories of a certain maximum length (this value is set at 10), possible in a gridworld. The number of states to be queried in `scott` is fixed at the maximum length of a trajectory possible (this value is set at 5 for `scott`). Also, with the increase in the size of the gridworld, the number of queries with `cst` increases. Further, we have not presented the number of queries for `rt` and `vft` in plots because they have well-defined mathematical formulae to calculate $|T|$.

## 8.2 Diagonal Actions

The performance summary for `rwt` and heuristics are presented in Table 4 and Figure 4. We observe similar trends to gridworld with smaller action space - the accuracy is high, and the false positive and false negative rates are extremely

Table 3: Different testers versus features sizes

| Tester | Feature size | Accuracy | False positive rate | False negative rate | Number of queries |
|---|---|---|---|---|---|
| rwt | 3 | 0.951± 0.051 | 0.037± 0.045 | 0.012± 0.034 | 1 |
| | 5 | 0.999± 0.004 | 0.001± 0.004 | 0.000± 0.002 | 1 |
| | 7 | 1.000± 0.001 | 0.000± 0.000 | 0.000± 0.001 | 1 |
| cst | 3 | 0.876± 0.097 | 0.000± 0.002 | 0.124± 0.097 | 31 |
| | 5 | 0.996± 0.007 | 0.000± 0.001 | 0.004± 0.007 | 29 |
| | 7 | 0.999± 0.002 | 0.000± 0.000 | 0.001± 0.002 | 28 |

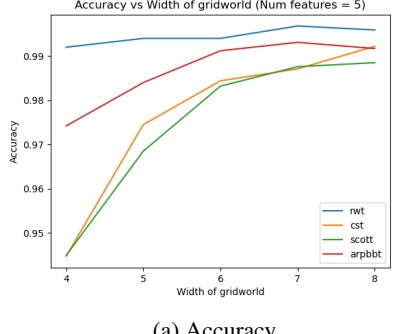

(a) Accuracy

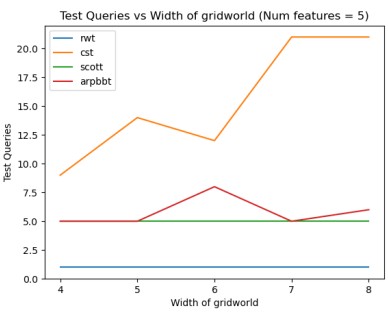

(b) Number of Test Queries

Figure 4: Tester performance for different gridworld widths with extended action space

Table 4: Different testers versus gridworld widths with extended action space

| Tester | Width | Accuracy | False positive rate | False negative rate | Number of queries |
|---|---|---|---|---|---|
| rwt | 4 | 0.992± 0.017 | 0.006± 0.015 | 0.003± 0.010 | 1 |
| | 6 | 0.994± 0.013 | 0.005± 0.011 | 0.001± 0.004 | 1 |
| | 8 | 0.996± 0.008 | 0.002± 0.005 | 0.002± 0.005 | 1 |
| cst | 4 | 0.945± 0.055 | 0.000± 0.001 | 0.055± 0.055 | 9 |
| | 6 | 0.984± 0.017 | 0.000± 0.001 | 0.016± 0.017 | 12 |
| | 8 | 0.992± 0.011 | 0.000± 0.000 | 0.008± 0.011 | 21 |

small, the number of queries with `cst` is higher than other heuristics, and the number of queries for `scott` is fixed at the maximum possible length of a trajectory. These results empirically indicate that the proposed testers are successfully able to verify the alignment of robots and humans in gridworlds with an extended action space.

### 8.3 Non-linear reward and state-feature relationships

The performance summary for `rwt` and `cst` is presented in Table 5 and Figure 5. We observe that for *cubic* relationship, the performance for both `rwt` and `cst` is close to that with *linear* relationship. Note that *cubic* approximates the *linear* relationship between $R$ and $w^T \phi(s)$, when $w^T \phi(s) \approx 0$. However, as expected for *exponential* relationship (assumption of Lemma 1 is no longer true), the performance for `rwt` is exceedingly poor while for `cst` the decline is negligible. This empirically enforces the importance and independence of linear relationship assumption between rewards and state features for exact value alignment algorithms (`rwt`) and heuristics (`cst`), respectively.

### 8.4 Critical States Tester with different thresholds

The performance of `cst` with different thresholds (0.0001 and 0.8, 0.2 is `cst` row in Table 2) is presented in Table 6. The corresponding figures are presented in Section 2 of the Supplementary Material. We observe that the accuracy for low threshold values is high whereas the accuracy drops considerably with higher threshold value. This is due to a decrease in the number of test queries with higher thresholds leading to a decrease in alignment verification ability. The

Table 5: Different testers versus gridworld widths with non-linear reward state-feature relationships

| Tester | Width | Accuracy | False positive rate | False negative rate | Number of queries |
|---|---|---|---|---|---|
| rwt (cubic) | 4 | 0.993± 0.013 | 0.004± 0.007 | 0.003± 0.011 | 1 |
| | 6 | 0.995± 0.008 | 0.003± 0.006 | 0.002± 0.005 | 1 |
| | 8 | 0.997± 0.006 | 0.001± 0.005 | 0.001± 0.004 | 1 |
| rwt (exponential) | 4 | 0.048± 0.052 | 0.953± 0.052 | 0.000± 0.000 | 1 |
| | 6 | 0.017± 0.021 | 0.983± 0.021 | 0.000± 0.000 | 1 |
| | 8 | 0.006± 0.012 | 0.994± 0.012 | 0.000± 0.000 | 1 |
| cst (cubic) | 4 | 0.968± 0.040 | 0.000± 0.000 | 0.032± 0.040 | 16 |
| | 6 | 0.991± 0.015 | 0.000± 0.000 | 0.010± 0.015 | 24 |
| | 8 | 0.995± 0.010 | 0.000± 0.000 | 0.005± 0.010 | 32 |
| cst (exponential) | 4 | 0.947± 0.051 | 0.000± 0.001 | 0.053± 0.051 | 16 |
| | 6 | 0.984± 0.022 | 0.000± 0.001 | 0.016± 0.022 | 16 |
| | 8 | 0.983± 0.099 | 0.010± 0.099 | 0.007± 0.010 | 31 |

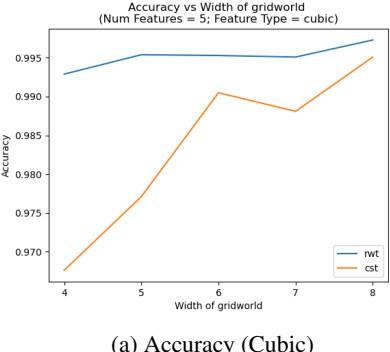

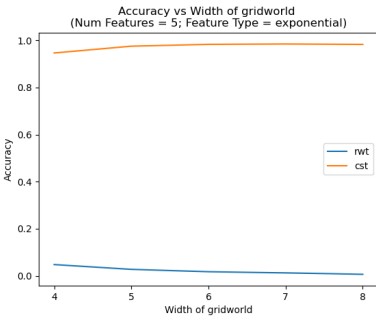

(a) Accuracy (Cubic)          (b) Accuracy (Exponential)

Figure 5: Tester performance for different reward - state features relationship

comparison between the number of test queries for different thresholds displays an expected trend, i.e., the number of states to be queried with lower thresholds is higher than those with a higher threshold.

Table 6: Critical states tester with different thresholds

| Tester | Width | Accuracy | False positive rate | False negative rate | Number of queries |
|---|---|---|---|---|---|
| cst (threshold = 0.0001) | 4 | 0.971± 0.036 | 0.000± 0.000 | 0.029± 0.036 | 16 |
| | 6 | 0.987± 0.018 | 0.000± 0.000 | 0.013± 0.018 | 24 |
| | 8 | 0.997± 0.007 | 0.000± 0.000 | 0.003± 0.007 | 32 |
| cst (threshold = 0.8) | 4 | 0.616± 0.447 | 0.362± 0.463 | 0.022± 0.032 | 1 |
| | 6 | 0.563± 0.482 | 0.431± 0.488 | 0.007± 0.013 | 4 |
| | 8 | 0.644± 0.468 | 0.354± 0.470 | 0.003± 0.008 | 3 |

## 9 Discussion

In this work, we implemented the algorithms and heuristics for Exact Value Alignment Verification. We observe that all the methods proposed in [7] can identify the alignment between a robot and a human agent with high confidence in two distinct scenarios, *implicit* and *explicit robot* with an *explicit human* agent. In this work, we have not investigated *implicit robot, implicit human* (approximate value alignment verification) setting due to lack of time. Additionally, we have carried out ablation studies to study the performance of these proposed methods in different settings, including an extended deterministic action space and non-linear reward state-feature relationship. Ultimately, a human agent could use any of the algorithms or heuristic (depending on the ability of the robot to access its rewards) to create a *driver's test* to test the robot's alignment.

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
