# OpenReview forum: "[Re]: Value Alignment Verification"
_ML_Reproducibility_Challenge/2021/Fall — RC2021_

### Official Review · Reviewer_hxST · 2022-02-18
**Looking good overall, but lacking an explicit comparison with the results and conclusions from the original paper**

**Rating:** 5
**Confidence:** 4

**Review:**

This report details the reproduction of the paper "Value Alignment Verification". Authors took the original code as inspiration but reimplemented the whole algorithm(s) from scratch. They verified that the proposed methods are indeed able to properly estimate value alignment with good accuracy on one of the two original environments (though they skipped one of the 3 settings from the paper, when there is no access to the human reward function). This report also investigates what happens (1) when additional (diagonal) actions are added to the grid world (=> the proposed approach still works), (2) when the reward is actually not linear in state features (=> this may break some algorithms, e.g., the Reward Weight Queries with exponential rewards), and (3) when the threshold for the Critical States method is varied (=> increasing it too much leads to fewer queries and lower accuracies)

Strengths:
* A lot of context is provided, making it possible to understand the key ideas and concepts behind the algorithms without having to read the original paper
* The code is well documented, and seems cleanly written (just skimming through it)
* A high amount of experimental results are reported

Weaknesses:
* The main concern I have (and the reason why I am not currently recommending acceptance) is that there is no explicit mention of how the empirical results provided in this report relate to those from the original paper. It's good to know that the algorithms seem to be working, but are the conclusions reached the same as those from the original paper? (in terms of comparisons between algorithms, and general trends). One specific potential concern (note that this is the first thing I noticed with a quick look, I haven't checked everything), when looking at Fig. 2a, compared to Fig. 6a in http://proceedings.mlr.press/v139/brown21a/brown21a-supp.pdf, is that the accuracy of the SCOT method is not as stable. I believe making links to the original results is a must have in such an empirical reproduction effort: just saying "the algorithm works" is not quite enough.
* The 3rd setting ("implicit human / implicit robot") hasn't been reproduced, and this is not mentioned in the "scope of reproducibility" paragraph.

Additional remarks:
* I would encourage the authors to provide additional details on the differences between their implementation vs the original one
* I didn't understand the last sentence at the bottom of p.3 ("Since the boundaries...")
* There's an extra parenthesis to be removed in the equation below l.117
* The argument for the exponential reward "breaking" rwt is the lack of linearity. I haven't fully thought this through, but another difference is also the fact that the reward doesn't go to zero when phi(x) is near zero, so I would have been curious to see if the same holds when subtracting 1 from the reward.
* l.216 "the algorithms for exact value alignment": please remind the reader which ones they are
* "Further, we have not presented the number of queries for rt and vft in plots because they have well-defined mathematical formulae to calculate |T |": this is not a convincing argument to not show them (it would still be interesting to see how they compare to other methods), also they do appear in Fig. 3b. In addition, not showing them in some figures like Fig. 2b is problematic because now the colors don't match those from Fig. 2a

---

### Official Review · Reviewer_V67A · 2022-03-29
**Review of [Re]: Value Alignment Verification**

**Rating:** 8
**Confidence:** 3

**Review:**

The authors attempted to reproduce the claims in the paper "Value Alignment Verification" by Daniel et al. 2021

- The reproducibility report is organized and well written.
- It was clearly written in the paper how the main claims of the original paper were reproduced.
- It was also clear how the authors implemented the algorithms and heuristics for Exact Value Alignment Verification.
- It was evident which algorithm was easier to reproduce, why, and what was challenging.
- The authors reached out to the original authors to validate some of their assumptions and clear doubts regarding the implementation.

---

### Meta-Review · Area_Chair_sajJ · 2022-04-09

**Recommendation:** Accept
**Confidence:** 3

**Metareview:**

Reviewers praised the code and writing, and the number of experiments.

---

### Decision · Program_Chairs · 2022-04-09

**Decision:**

Accept

**Comment:**

Following the recommendation of reviewers and meta-reviewer, the paper is accepted for ML Reproducibility Challenge 2021, and will be published in the upcoming special edition of ReScience Journal.